# Explaining Naive Bayes and Other Linear Classifiers with Polynomial Time and Delay

**Joao Marques-Silva**[1], **Thomas Gerspacher**[2], **Martin C. Cooper**[1]
[1]IRIT, CNRS, University of Toulouse III, France
[2]ANITI, University of Toulouse, France
`{joao.marques-silva,thomas.gerspacher,cooper}@irit.fr`

**Alexey Ignatiev**
Monash University, Australia
`alexey.ignatiev@monash.edu`

**Nina Narodytska**
VMware Research, CA, USA
`nnarodytska@vmware.com`

## Abstract

Recent work proposed the computation of so-called PI-explanations of Naive Bayes Classifiers (NBCs) [35]. PI-explanations are subset-minimal sets of feature-value pairs that are sufficient for the prediction, and have been computed with state-of-the-art exact algorithms that are worst-case exponential in time and space. In contrast, we show that the computation of one PI-explanation for an NBC can be achieved in log-linear time, and that the same result also applies to the more general class of linear classifiers. Furthermore, we show that the enumeration of PI-explanations can be obtained with polynomial delay. Experimental results demonstrate the performance gains of the new algorithms when compared with earlier work. The experimental results also investigate ways to measure the quality of heuristic explanations.

## 1 Introduction

Approaches proposed in recent years for computing explanations of Machine Learning (ML) models can be broadly characterized as *heuristic* or *non-heuristic*[1]. Heuristic approaches denote those providing *no* formal guarantees on their results. In contrast, non-heuristic approaches *do* provide some sort of formal guarantee(s) on their results, usually at the cost of increased computational complexity. Among the heuristic approaches for finding explanations, two have been studied in greater detail. One line of work focuses on devising model-agnostic linear approximations of the underlying model [29, 18]. Another line of work is exemplified by Anchor [30], and targets the computation of a set of feature-value pairs associated with a given instance as a way of explaining the prediction. To date, all non-heuristic methods have focused on computing sets of feature-value pairs that are sufficient for the prediction [35, 10, 36, 6][2]. Moreover, in terms of formal guarantees, [35] studies two distinct definitions of explanations. A PI-explanation represents a subset-minimal set of feature values that entails the outcome of the decision function for the predicted class whatever the values of the other features (i.e. it represents a *prime implicant* of the outcome of the decision function). PI-explanations have also been studied under the name of abductive explanations [10]. In contrast, and assuming binary features, an MC-explanation is a cardinality-minimal set of equal-valued features that entails the outcome of the decision function. Non-heuristic approaches are model-based, and

so earlier work specifically considered Naive-Bayes Classifiers (NBCs) and Latent-Tree Classifiers (LTCs) [35, 6], Bayesian Network Classifiers [36, 6], and Neural Networks [10, 34].

In the concrete case of computing (non-heuristic) PI-explanations for NBCs, earlier work [35] proposed algorithms that are worst-case exponential in both time and space. In contrast, in this paper we propose a novel non-heuristic solution for computing PI-explanations of NBCs and other linear classifiers [3], which exhibits two fundamental advantages over earlier work. First, the paper shows that computing PI-explanations for NBCs (but also for any linear classifier) is in P, by proposing a log-linear algorithm for computing one smallest size PI-explanation. Second, the paper proposes a polynomial (log-linear) delay algorithm for enumerating the PI-explanations of NBCs (and also of any linear classifier). Furthermore, the paper presents an experimental evaluation of different approaches for explaining NBCs with PI-explanations, including the heuristic solutions computed by Anchor [30] and SHAP [18][4]. Moreover, although (real-valued) linear classifiers can be viewed as interpretable [29], this does not equate with computing PI-explanations, particularly when features are categorical. To the best of our knowledge, proving the (polynomial) complexity of computing PI-explanations for linear classifiers (including NBCs) closes an open problem. Furthermore, the results in this paper rank among the first to investigate classes of ML models for which PI-explanations can be computed in polynomial time [2, 11].

**Related work.** In a recent paper [34], Shi *et al.* investigate the use of knowledge compilation in the analysis of machine learning models. The main focus of the paper is on compilation of binarized neural networks. However, the authors consider other ML models, including linear classifiers. In particular, the authors state that a linear classifier with integer weights can be compiled into an OBDD in pseudo-polynomial time $O(nW)$, where $W$ is the sum of weights in the linear classifier. These results are interesting and complement our results in terms of providing evidence that XLCs form a tractable class of ML models. Nevertheless, our algorithm does not depend on $W$ and runs in log-linear time which is more efficient than [34] for finding and enumerating PI-explanations.

Another line of research focuses on designing algorithms for explaining Bayesian networks [40, 42, 21, 16, 39]. For example, Mengshoel *et al.* investigate the problem of finding the most probable explanation in Bayesian networks. The authors encode the problem as a MaxSAT problem and employ stochastic local search to find the most probable explanation. Another example is [39], where the authors use scenarios for constructing and understanding a Bayesian network for legal evidence and argue that this form of explanation is useful for criminal trial judges. Overall, these approaches propose several definitions of explanation and a wide range of algorithms to compute them for BNs. Note that we focus on NBCs which is a special class of BN and exploit the properties of NBCs to design a polynomial algorithm for finding a PI-explanation. Finally, [40] investigates the the concept of explaining away as a form of intercausal reasoning.

**Organization.** The paper is organized as follows. Section 2 introduces the concepts and notation used throughout the paper. Section 3 introduces XLCs (a simple extension of linear classifiers (LCs)), and develops a new approach for computing, in polynomial time, one PI-explanation for XLCs. Section 3 also proposes a polynomial delay algorithm for the enumeration of PI-explanations of XLCs. Section 4 compares dedicated approaches for explaining NBCs [35] with the algorithms proposed in this paper, but also with the explanations produced by heuristic approaches. The paper concludes in Section 5.

## 2 Preliminaries

**Explanations of ML models.** We consider a classification problem with two classes $\mathcal{K} = \{\oplus, \ominus\}$, defined on a set of features (or attributes) $e_1, \ldots, e_n$, which will be represented by their indices $\mathcal{E} = \{1, \ldots, n\}$. The features can either be real-valued or categorical. For real-valued features, we have $\lambda_i \leq e_i \leq \mu_i$, where $\lambda_i, \mu_i$ are given lower and upper bounds. For categorical features, we

have $e_i \in \{1, \ldots, d_i\}$. A concrete assignment to the features referenced by $\mathcal{E}$ is represented by an $n$-dimensional vector $\mathbf{a} = (a_1, \ldots, a_n)$, where $a_j$ denotes the value assigned to feature $j$, represented by variable $e_j$, such that $a_j$ is taken from the domain of $e_j$. The set of all $n$-dimensional vectors denotes the *feature space* $\mathbb{E}$. Given a classifier with features $\mathcal{E}$, a *decision function* [35] is a mapping from the feature space to the set of classes, i.e. $\tau : \mathbb{E} \to \mathcal{K}$. For example, for a linear classifier, the decision function picks $\oplus$ if $\sum_i w_i e_i > 0$, and $\ominus$ if $\sum_i w_i e_i \leq 0$. Given $\mathbf{a} \in \mathbb{E}$, with $\tau(\mathbf{a}) = \oplus$, we consider the set of feature literals of the form $(e_i = a_i)$, where $e_i$ denotes a variable and $a_i$ a constant. A PI-explanation [35] is a subset-minimal set $\mathcal{P} \subseteq \mathcal{E}$, denoting feature literals, such that

$$\forall (\mathbf{e} \in \mathbb{E}). \bigwedge\nolimits_{j \in \mathcal{P}} (e_j = a_j) \to \tau(\mathbf{e}) = \oplus \tag{1}$$

is true. Alternatively, we can represent (1) as a rule:

$$\textbf{IF} \quad \bigwedge\nolimits_{j \in \mathcal{P}} (e_j = a_j) \quad \textbf{THEN} \quad \tau(\mathbf{e}) = \oplus. \tag{2}$$

(The same definitions apply in the case of class $\ominus$ (given $\mathbf{a} \in \mathbb{E}$, with $\tau(\mathbf{a}) = \ominus$).)

**Naive Bayes Classifier (NBC).** NBCs [7] can be viewed as special cases of Bayesian Network Classifiers (BNCs) [8], that make strong conditional independence assumptions among the features. Graphically, NBCs are represented as depicted in Figure 1 for a concrete example. Given some evidence $\mathbf{e}$ (in our case, this is an assignment to the features), the predicted class is given by:

$$\tau(\mathbf{e}) = \operatorname{argmax}_{c \in \mathcal{K}} \left( \Pr(c|\mathbf{e}) \right). \tag{3}$$

It is well known that $\Pr(c|\mathbf{e})$ can be computed as follows: $\Pr(c|\mathbf{e}) = \frac{\Pr(c, \mathbf{e})}{\Pr(\mathbf{e})}$. However, $\Pr(\mathbf{e})$ is constant for every $c \in \mathcal{K}$. Hence, (3) can be rewritten as follows:

$$\tau(\mathbf{e}) = \operatorname{argmax}_{c \in \mathcal{K}} \left( \Pr(c, \mathbf{e}) \right) \tag{4}$$

Finally, assuming features to be mutually conditional independent, (4) can be rewritten as follows:

$$\tau(\mathbf{e}) = \operatorname{argmax}_{c \in \mathcal{K}} \left( \Pr(c) \times \prod\nolimits_i \Pr(e_i|c) \right). \tag{5}$$

A standard transformation is to apply logarithms, thus getting:

$$\tau(\mathbf{e}) = \operatorname{argmax}_{c \in \mathcal{K}} \left( \log \Pr(c) + \sum\nolimits_i \log \Pr(e_i|c) \right). \tag{6}$$

Also, if $\Pr(e_i|c) = 0$, then we use instead a sufficiently large negative value $\mathbb{M}$ [27] [5], i.e. we pick $\max(\mathbb{M}, \log(\Pr(e_i|c))) \in [\mathbb{M}, 0]$. (A simple solution is to use the sum of the logarithms of all the non-zero probabilities plus some $\epsilon < 0$.) For simplicity, i.e. to work with positive values, we can add a sufficiently large positive threshold $\mathbb{T}$ to each probability, to serve as a reference, thus obtaining:

$$\tau(\mathbf{e}) = \operatorname{argmax}_{c \in \mathcal{K}} \left( (\mathbb{T} + \log \Pr(c)) + \sum\nolimits_i (\mathbb{T} + \log \Pr(e_i|c)) \right). \tag{7}$$

(For example, we can set $\mathbb{T}$ to the complement of the negative value with the largest absolute value.) Also for simplicity, we use the notation $\operatorname{lPr}(\alpha) \triangleq \mathbb{T} + \max(\mathbb{M}, \log(\Pr(\alpha)))$.

**Running Example.** Consider the NBC shown in Figure 1 [6]. The features are the random variables $R_1$, $R_2$, $R_3$ and $R_4$. Each $R_i$ can take values $\mathbf{t}$ or $\mathbf{f}$ denoting, respectively, whether a listener likes or not that radio station. Random variable $G$ denotes an age class, which can take values $\mathsf{Y}$ and $\mathsf{O}$, denoting young and older listeners, respectively. Using the notation proposed earlier, we will use $\oplus$ for $\mathsf{Y}$ and $\ominus$ for $\mathsf{O}$. We also associate $\oplus$ with 1 or $\mathbf{t}$ and $\ominus$ with 0 or $\mathbf{f}$. In general we have

$$\Pr(G, R_1, R_2, R_3, R_4) = \Pr(G) \times \Pr(R_1|G) \times \Pr(R_2|G) \times \Pr(R_3|G) \times \Pr(R_4|G). \tag{8}$$

Considering the assignment $(G, R_1, R_2, R_2, R_3) = (\oplus, \mathbf{t}, \mathbf{f}, \mathbf{t}, \mathbf{f})$, and using $g$ to denote $G = \oplus$, $r_i$ to denote $R_i = \mathbf{t}$ and $\neg r_i$ to denote $R_i = \mathbf{f}$, (8) can be written as follows:

$$\Pr(g, r_1, \neg r_2, r_3, \neg r_4) = \Pr(g) \times \Pr(r_1|g) \times \Pr(\neg r_2|g) \times \Pr(r_3|g) \times \Pr(\neg r_4|g).$$

Let us consider $\mathbf{a} = (R_1, R_2, R_3, R_4) = (\mathbf{t}, \mathbf{f}, \mathbf{t}, \mathbf{f})$. Since all probabilities are strictly positive, we set $\mathbb{M}$ to a very large negative (irrelevant) value. In addition, we set $\mathbb{T}$ to a value above the complement of the logarithm of the smallest probability (i.e. 0.02), e.g we can set $\mathbb{T} = 4 > -\log(0.02)$. Using (7), we get the values shown in Figure 2. As can be concluded, the prediction will be $\oplus$. Observe that neither the value of $\mathbb{M}$ nor of $\mathbb{T}$ affect the prediction.

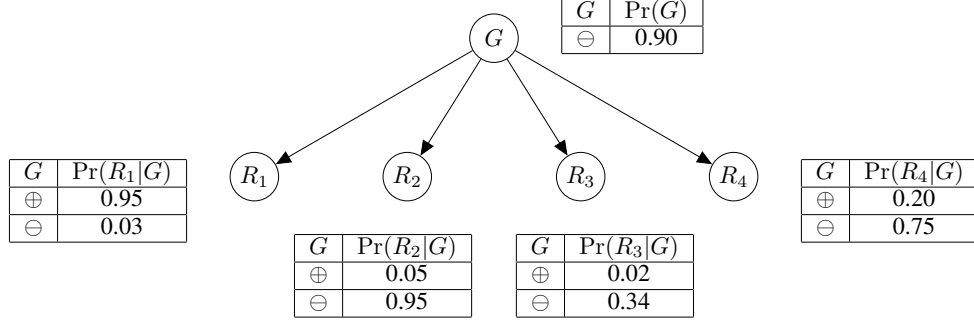

Figure 1: Running example.

|            | $\Pr(g)$ | $\Pr(r_1|g)$ | $\Pr(\neg r_2|g)$ | $\Pr(r_3|g)$ | $\Pr(\neg r_4|g)$ | $\text{lPr}(\oplus|\mathbf{a})$ |
|------------|----------|--------------|-------------------|--------------|-------------------|---------------------------------|
| $\Pr(\cdot)$ | 0.10 | 0.95 | 0.95 | 0.02 | 0.80 | |
| $\text{lPr}(\cdot)$ | 1.70 | 3.95 | 3.95 | 0.09 | 3.78 | 13.47 |

(a) Computing $\text{lPr}(\oplus|\mathbf{a})$

|            | $\Pr(\neg g)$ | $\Pr(r_1|\neg g)$ | $\Pr(\neg r_2|\neg g)$ | $\Pr(r_3|\neg g)$ | $\Pr(\neg r_4|\neg g)$ | $\text{lPr}(\ominus|\mathbf{a})$ |
|------------|---------------|-------------------|------------------------|-------------------|------------------------|----------------------------------|
| $\Pr(\cdot)$ | 0.90 | 0.03 | 0.05 | 0.34 | 0.25 | |
| $\text{lPr}(\cdot)$ | 3.89 | 0.49 | 1.00 | 2.92 | 2.61 | 10.91 |

(b) Computing $\text{lPr}(\ominus|\mathbf{a})$

Figure 2: Deciding prediction for $\mathbf{a} = (\mathbf{t}, \mathbf{f}, \mathbf{t}, \mathbf{f})$

## 3 Explaining Extended Linear Classifiers

This section first introduces Extended Linear Classifiers (XLCs) and then details how PI-explanations can be computed for predictions of XLCs.

### 3.1 Extended Linear Classifiers

Let $\mathcal{E}$ be partitioned into $\mathcal{R}$ and $\mathcal{C}$, denoting respectively the real-valued and the categorical features. Each real-valued feature with index $i \in \mathcal{R}$ takes bounded values $\lambda_i \le e_i \le \mu_i$. For each categorical feature $j \in \mathcal{C}$, $e_j \in \{1, \ldots, d_j\}$.

We consider an XLC, that encompasses real-valued and categorical features. Let

$$\nu(\mathbf{e}) \triangleq w_0 + \sum_{i \in \mathcal{R}} w_i e_i + \sum_{j \in \mathcal{C}} \sigma(e_j, v_j^1, v_j^2, \ldots, v_j^{d_j}), \tag{9}$$

where $\sigma$ is a selector function that picks the value $v_j^r$ iff $e_j$ takes value $r$. Moreover, let us define the decision function, $\tau(\mathbf{e}) = \oplus$ if $\nu(\mathbf{e}) > 0$ and $\tau(\mathbf{e}) = \ominus$ if $\nu(\mathbf{e}) \le 0$.

**Reducing linear classifiers to XLCs.** For a linear classifier, with only real-valued features, simply set $\mathcal{C} = \emptyset$. For an NBC with boolean features[7], we consider a different reduction with $\mathcal{R} = \emptyset$, starting from (7). Moreover, the $\operatorname{argmax}$ operator in (7) can be replaced by an inequality, from which we get

$$\text{lPr}(\oplus) - \text{lPr}(\ominus) + \sum_{i=1}^n (\text{lPr}(e_i|\oplus) - \text{lPr}(e_i|\ominus))e_i + \sum_{i=1}^n (\text{lPr}(\neg e_i|\oplus) - \text{lPr}(\neg e_i|\ominus))\neg e_i > 0 \tag{10}$$

The reduction is completed by setting: $w_0 \triangleq \text{lPr}(\oplus) - \text{lPr}(\ominus)$, $v_j^1 \triangleq \text{lPr}(\neg e_j|\oplus) - \text{lPr}(\neg e_j|\ominus)$, $v_j^2 \triangleq \text{lPr}(e_j|\oplus) - \text{lPr}(e_j|\ominus)$, and $d_j \triangleq 2$.

*Example* 1. Figure 3a shows the resulting XLC formulation for the example in Figure 2. We also let $\mathbf{f}$ be associated with value 1 and $\mathbf{t}$ be associated with value 2, and $d_j = 2$.

### 3.2 Explaining XLCs

We now investigate how (smallest or cardinality-minimal) PI-explanations can be computed for XLCs, and also how (minimal) PI-explanations can be enumerated. First, we show that the problem

| $w_0$ | $v_1^1$ | $v_1^2$ | $v_2^1$ | $v_2^2$ | $v_3^1$ | $v_3^2$ | $v_4^1$ | $v_4^2$ |
|---|---|---|---|---|---|---|---|---|
| -2.19 | -2.97 | 3.46 | 2.95 | -2.95 | 0.4 | -2.83 | 1.17 | -1.32 |

| $\Gamma$ | $\delta_1$ | $\delta_2$ | $\delta_3$ | $\delta_4$ | $\Phi$ |
|---|---|---|---|---|---|
| 2.56 | 6.43 | 5.90 | 0.00 | 2.49 | 12.26 |

(a) Example reduction of NBC to XLC (Example 1)  (b) Computing $\delta_j$'s for the XLC (Example 2)

Figure 3: Values used in the running example (Example 1 and Example 2)

of computing PI-explanations for XLCs can be reduced to a variant of the knapsack problem. We demonstrate that this class of knapsack problems can be solved in log-linear time. Second, we exploit special properties of our reformulation and derive an efficient enumeration algorithm for PI-explanations. Finally, we prove correctness of the proposed algorithms.

We start by assessing how *free* some of the features are. For a given instance $\mathbf{e} = \mathbf{a}$, define a *constant slack* (or gap) value $\Gamma$ given by

$$\Gamma^a \triangleq \nu(\mathbf{a}) = w_0 + \sum\nolimits_{i \in \mathcal{R}} w_i a_i + \sum\nolimits_{j \in \mathcal{C}} \sigma(a_j, v_j^1, v_j^2, \ldots, v_i^{d_j}), \tag{11}$$

i.e. this is the value obtained when deciding $\oplus$ to be the picked class, given the assignment $\mathbf{e} = \mathbf{a}$.

We are interested in computing one PI-explanation [35] of an XLC, but we are also interested in enumerating PI-explanations. As argued in Section 2, this corresponds to finding a subset-minimal set of literals $\mathcal{P} \subseteq \mathcal{E}$ such that (1) holds, or alternatively

$$\forall (\mathbf{e} \in \mathbb{E}). \bigwedge\nolimits_{j \in \mathcal{P}} (e_j = a_j) \rightarrow (\nu(\mathbf{e}) > 0), \tag{12}$$

assuming that $\nu(\mathbf{a}) > 0$. In what follows, we partition $\mathcal{E}$ into $\mathcal{P}$ and $\mathcal{N}$, respectively the picked and the non-picked attributes from $\mathcal{E}$.

**Categorical case.** Let us first consider $\mathcal{R} = \emptyset$. Each feature $e_j$ is assigned value $a_j$, which results in selecting some value $v_j^{a_j}$, i.e. the value from the weights associated with $e_j$ which is picked when $e_j = a_j$. Thus, $\Gamma$ is computed as follows: $\Gamma^a = w_0 + \sum_{j \in \mathcal{C}} v_j^{a_j}$.

Moreover, let $v_j^\omega$ denote the *smallest* (or *worst-case*) value associated with $e_j$. Then, by letting every $e_j$ take *any* value, the *worst-case* value of $\nu(\mathbf{e})$ is

$$\Gamma^\omega = w_0 + \sum\nolimits_{j \in \mathcal{C}} v_j^\omega. \tag{13}$$

We are interested in cases where $\Gamma^\omega \leq 0$, corresponding to predicting $\ominus$ instead of $\oplus$. (Otherwise the prediction would not change from $\oplus$.) The expression above can be rewritten as follows

$$\begin{aligned} \Gamma^\omega &= w_0 + \sum_{j \in \mathcal{C}} v_j^{a_j} - \sum_{j \in \mathcal{C}} (v_j^{a_j} - v_j^\omega) \\ &= \Gamma^a - \sum_{j \in \mathcal{C}} \delta_j = -\Phi, \end{aligned} \tag{14}$$

where we use $\delta_j \triangleq v_j^{a_j} - v_j^\omega$, and $\Phi \triangleq \sum_{j \in \mathcal{C}} \delta_j - \Gamma^a = -\Gamma^\omega$. Our goal is to find a smallest (or subset-minimal) set $\mathcal{P}$ such that the prediction is still $\oplus$ (whatever the values of the other features):

$$w_0 + \sum\nolimits_{j \in \mathcal{P}} v_j^{a_j} + \sum\nolimits_{j \notin \mathcal{P}} v_j^\omega = -\Phi + \sum\nolimits_{j \in \mathcal{P}} \delta_j > 0, \tag{15}$$

i.e. we want to pick a smallest (or subset-minimal) set of literals that ensures that the prediction will be $\oplus$. In turn, (15) can be represented as the following optimization problem:

$$\begin{aligned} \min \quad & \sum_{i=1}^n p_i \\ \text{s.t.} \quad & \sum_{i=1}^n \delta_i p_i > \Phi \\ & p_i \in \{0, 1\}, \end{aligned} \tag{16}$$

where the variables $p_i$ assigned value 1 denote the indices included in $\mathcal{P}$. Although solving (16) seems to equate to solving an NP-hard optimization, concretely the minimization version of the knapsack problem [14], the fact that the coefficients in the cost function are all equal to 1 makes the problem solvable in log-linear time[8]. Concretely, we can now develop a greedy algorithm that

**Function** ONEEXPLANATION ($\mathsf{Vs},\mathsf{Flip},\Delta,\Phi^R,\mathsf{Idx},\mathsf{Xpl}$) ;

    **Input:** $\mathsf{Vs}$: Values of instance being explained; $\mathsf{Flip}$: Array reference of decision steps;
        $\Delta$: Sorted $\delta_j$'s; $\Phi^R$: Explanation threshold; $\mathsf{Idx}$: Index for $\Delta$; $\mathsf{Xpl}$: Set reference of
        explanation literals

    **Output:** $\Phi^R$: Updated threshold; $\mathsf{Idx}$: Updated index for $\Delta$

1     **while** $\Phi^R \geq 0$ **do**
2         $\mathsf{Idx} \leftarrow \mathsf{Idx} + 1$ ;
3         $\mathsf{Flip}[\mathsf{Idx}] \leftarrow 0$ ;
4         $\Phi^R \leftarrow \Phi^R - \Delta[\mathsf{Idx}]$ ;
5         $\mathsf{Xpl} \leftarrow \mathsf{Xpl} \cup \{(e_{\mathsf{Idx}}, \mathsf{Vs}[\mathsf{Idx}])\}$ ;
6     REPORTEXPLANATION ($\mathsf{Xpl}$) ;
7     **return** ($\Phi^R, \mathsf{Idx}$) ;

**Algorithm 1:** Finding one explanation

computes a smallest PI-explanation, representing one optimal solution of (16). At each step, we simply pick the largest $\delta_i$ that has not yet been picked. Optimality of the computed solution is given by Proposition 1 [9].

**Proposition 1.** *Let $\mathcal{S} = \langle l_1, \ldots, l_n \rangle$ represent indices of $\mathcal{E}$ sorted by non-increasing value of $\delta_j$. Pick $k$ such that $\sum_{j \in \{l_1, \ldots, l_k\}} \delta_j > \Phi$ and $\sum_{j \in \{l_1, \ldots, l_{k-1}\}} \delta_j \leq \Phi$. Then (12) holds for $\mathcal{P} = \{p_{l_r} | 1 \leq r \leq k\}$, and $\mathcal{P}$ represents an optimal solution of (16).*

*Example* 2. Figure 3b shows the values used for computing explanations for the example in Figure 2. For this example, the sorted $\delta_j$'s become $\langle \delta_1, \delta_2, \delta_4, \delta_3 \rangle$. By picking $\delta_1$ and $\delta_2$, we ensure that the prediction is $\oplus$, independently of the evidence provided for features $e_3$ and $e_4$. Thus $(e_1) \wedge (\neg e_2)$ is a PI-explanation for the NBC shown in Figure 1, with evidence $(e_1, e_2, e_3, e_4) = (\mathbf{t}, \mathbf{f}, \mathbf{t}, \mathbf{f})$. (It is easy to observe that $\tau(\mathbf{t}, \mathbf{f}, \mathbf{f}, \mathbf{f}) = \tau(\mathbf{t}, \mathbf{f}, \mathbf{f}, \mathbf{t}) = \tau(\mathbf{t}, \mathbf{f}, \mathbf{t}, \mathbf{f}) = \tau(\mathbf{t}, \mathbf{f}, \mathbf{t}, \mathbf{t}) = \oplus$.)

In the concrete case of NBCs, if the goal is to compute a single explanation, then the algorithm detailed in this section is exponentially more efficient (in the worst case) than earlier work [35]. However, in some settings one wants to be able to analyze some or even all explanations for a given instance (this is further discussed in Section 4). We describe next a polynomial (log-linear) delay algorithm for enumeration of explanations for XLCs (and so for NBCs).

**Enumerating explanations with polynomial delay.** As shown above, a smallest PI-explanation can be computed in log-linear time by sorting the $\delta_i$ values and picking the first $k$ literals that ensure the prediction. We start by presenting a more elaborate description of the algorithm, which we then use for devising the enumeration of explanations with polynomial delay [10]. Algorithm 1 shows the pseudo-code for computing one smallest explanation. $\Delta$ denotes the array of sorted $\delta_j$'s. (The pseudo-code assumes that the order $1, 2, \ldots, n$ represents the literals in sorted order.) $\Phi^R$ is initialized with the value of $\Phi$, being updated as the algorithm(s) progress(es). Algorithm 1 corresponds to the direct application of Proposition 1. This algorithm can now be exploited for implementing a polynomial delay algorithm for enumerating PI-explanations. Algorithm 2 depicts the enumeration of PI-explanations. The algorithm implements a (restricted) backtrack search procedure, which in some circumstances can be shown to yield polynomial delay algorithms [4]. $\mathsf{Idx}$ denotes the depth of the search tree and $\mathsf{Flip}$ (if assigned 0) records which $\delta_j$'s are used for updating $\Phi^R$. (The entries of $\mathsf{Flip}$ take value -1 if unused, and value 1 if have been backtracked upon.) A key aspect of the algorithm is that it only branches when it is guaranteed that a PI-explanation can still be found, given the prefix (of picked or not picked $\delta_j$'s) defined by $\mathsf{Flip}$ and $\mathsf{Idx}$. Otherwise, the algorithm must backtrack and enter a consistent state (with at most a linear backtracking effort). Algorithm 3 shows the backtrack step of the PI-enumeration algorithm. Algorithm 3 terminates if no more PI-explanations can be found, or with the guarantee that another PI-explanation can be extracted with Algorithm 1. It is straightforward to conclude that both Algorithm 1 and Algorithm 3 run in linear time on the size of the current depth of the search tree (which is linear on the number of features). Thus, we can list PI-explanations of XLC's with polynomial delay [11].

**Function** ALLEXPLANATIONS($\mathsf{Vs}, \Delta, \Phi^R$) ;

    **Input:** $\mathsf{Vs}$: Values of instance being explained; $\Delta$: Sorted $\delta_j$'s; $\Phi^R$: Explanation threshold

1    $(\mathsf{Xpl}, \mathsf{Flip}, \mathsf{Idx}) \leftarrow (\emptyset, [-1, \ldots, -1], 0)$ ;

2    **while** $\mathsf{Idx} \geq 0$ **do**

3        $(\Phi^R, \mathsf{Idx}) \leftarrow$ ONEEXPLANATION($\mathsf{Vs}, \mathsf{Flip}, \Delta, \Phi^R, \mathsf{Idx}, \mathsf{Xpl}$) ;

4        $(\Phi^R, \mathsf{Idx}) \leftarrow$ ENTERVALIDSTATE($\mathsf{Vs}, \mathsf{Flip}, \Delta, \Phi^R, \mathsf{Idx}, \mathsf{Xpl}$) ;

**Algorithm 2:** Finding all explanations

**Function** ENTERVALIDSTATE($\mathsf{Vs}, \mathsf{Flip}, \Delta, \Phi^R, \mathsf{Idx}, \mathsf{Xpl}$) ;

    **Input:** $\mathsf{Vs}$: Values of instance being explained; $\mathsf{Flip}$: Array reference of decision steps;
        $\Delta$: Sorted $\delta_j$'s; $\Phi^R$: Explanation threshold; $\mathsf{Idx}$: Index for $\Delta$; $\mathsf{Xpl}$: Set reference of
        explanation literals

    **Output:** $\Phi^R$: Updated threshold; $\mathsf{Idx}$: Updated index for $\Delta$

1    **while** $\Phi^R < 0$ **or** $\sum_{i=\mathsf{Idx}}^{n} \Delta[i] < \Phi^R$ **do**

2        **while** $\mathsf{Idx} \geq 0 \wedge \mathsf{Flip}[\mathsf{Idx}] = 1$ **do**

3            $\mathsf{Flip}[\mathsf{Idx}] \leftarrow -1$ ;

4            $\mathsf{Idx} \leftarrow \mathsf{Idx} - 1$ ;

5        **if** $\mathsf{Idx} < 0$ **then return** $(\Phi^R, \mathsf{Idx})$ ;

6        $\mathsf{Xpl} \leftarrow \mathsf{Xpl} \setminus \{(e_{\mathsf{Idx}}, \mathsf{Vs}[\mathsf{Idx}])\}$;

7        $\Phi^R \leftarrow \Phi^R + \Delta[\mathsf{Idx}]$;

8        $\mathsf{Flip}[\mathsf{Idx}] \leftarrow 1$;

9    **return** $(\Phi^R, \mathsf{Idx})$ ;

**Algorithm 3:** Entering a valid state

**Proposition 2.** *PI-explanations of an XLC can be enumerated with log-linear delay.*

**Real-valued & mixed case.** Let us now consider $\mathcal{R} \neq \emptyset$. As before, the prediction is assumed to be $\oplus$. For each feature, if $w_i > 0$, then we are interested in assessing the impact of reducing the value of $e_i$. Hence, the worst-case scenario is achieved when $e_i = \lambda_i$. In this case, we define $\delta_i = (a_i - \lambda_i)w_i$. A no-change constraint on the value of $e_i$ is formulated as $e_i \geq a_i$ (i.e. we *clamp* the value of $e_i$ by imposing a lower bound on its value). In contrast, if $w_i < 0$, then we are interested in assessing the impact of increasing the value of $e_i$. The worst-case scenario is now $e_i = \mu_i$. In this case, we define $\delta_i = (a_i - \mu_i)w_i$. Moreover, a no-change constraint on the value of $e_i$ is formulated as $e_i \leq a_i$ (i.e. in this case we *clamp* the value of $e_i$ by imposing an upper bound on its value). Given the definition of the $\delta_i$ constants for real-valued features, and associated literals in case of a no-change constraint, we can compute explanations using the restricted knapsack problem formulation as above. Thus, we can also compute one cardinality optimal solution in log-linear time, and enumerate subset-minimal solutions with polynomial delay.

## 4  Experimental Evaluation

This section evaluates the PI-explanation enumerator XPXLC, that implements the algorithms described in this paper[12]. XPXLC was tested in Debian Linux on an Intel Xeon CPU 5160 3.00 GHz with 64 GByte of memory. When testing scalability, XPXLC was run with 8GByte limit on RAM and two hours time limit. The experiment was divided into 3 parts: (1) evaluating the raw performance of XPXLC, (2) comparing it with the state-of-the-art compilation approach STEP [35, 36], and (3) using complete enumeration of PI-explanations to assess the quality of explanations of the well-known heuristic explainers Anchor [30] and SHAP [18].

**Datasets.** We selected a set of widely-used, publicly available, datasets from [37, 28, 13]. The total number of datasets used is 37. These datasets contain tabular data with up to 32 features per dataset. The task is to perform classification. For each dataset, we trained a Naive Bayes classifier[13] using 80% of the training data. The average test accuracy assessed for the 20% remaining instances is 77.7%.

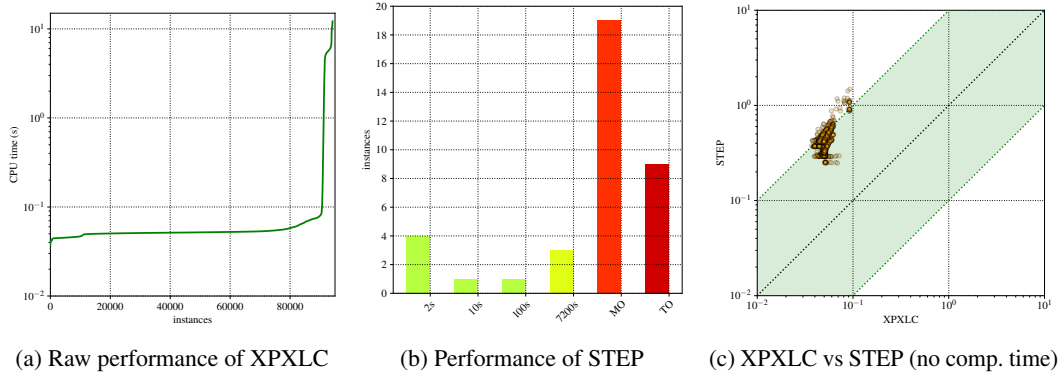

(a) Raw performance of XPXLC    (b) Performance of STEP    (c) XPXLC vs STEP (no comp. time)

Figure 4: Scalability of XPXLC targeting $10^6$ PI-explanations, performance of STEP, and comparative performance of XPXLC and STEP.

(All the datasets and the trained classifiers are available in the online repository.) The experiments targeted XPXLC's ability to enumerate a given number of explanations within a time limit.

**Raw performance.** Figure 4a shows the scalability of XPXLC. Here, XPXLC was set to compute $10^6$ distinct explanations for each instance of each dataset. For the cases having fewer than $10^6$ explanations, XPXLC terminates as soon as all explanations are computed. The smallest number of observed explanations per instance is 1, the maximum number is at least $10^6$, while on average 29207.5 PI-explanations are reported per each instance. The total number of instances to explain in this experiment is 94174. The line drawn through point $(x, y)$ in Figure 4a shows how many instances on the $X$-axis are solved by the time shown on the $Y$-axis. As can be observed, performance is not an issue for XPXLC – it never exceeds 12 seconds to enumerate $10^6$ explanations for each of the target instances. On average, XPXLC finishes complete enumeration (of at most $10^6$ explanations) in 0.23 seconds.

**Enumerative vs. compilation-based approaches.** The state of the art for finding PI-explanations for NBCs is the STEP compilation-based approach [35, 36, 38]. Concretely, STEP consists of (1) compilation of a BNC classifier into a *sentential decision diagram* (SDD) and (2) enumeration of PI-explanations using efficient algorithms for SDD-based prime implicant enumeration. The existing implementation of STEP can only handle binary features. Therefore, and in order to compare the relative performance of XPXLC and STEP, we apply a one-hot encoding (OHE) to categorical features, retrain the Naive Bayes classifiers and run both tools on the OHE instances[14], targeting the complete enumeration of explanations. Moreover, despite its worst-case exponential complexity in time and space, STEP can still compile into SDDs 9 (out of 37) NBC classifiers, i.e. close to 25% of the classifiers, within the 2 hours time limit and 32 GByte memory limit. Once an NBC classifier is compiled into an SDD, enumeration of all PI-explanations is relatively easy — concretely, it takes 0.39 seconds for the compilation-based approach to enumerate all explanations. However, the SDD compilation step itself takes between 1 and 4300 seconds for the classifiers that can be compiled. If the compilation time is amortized over all data instances of each dataset, its impact ranges from a fraction of a second to ≈50 seconds. Figure 4b shows a histogram summarizing the performance of STEP's compiler. The bars in the histogram represent the classifiers that STEP is able to compile within 2 seconds (there are 4 of them), 10 seconds (1), 100 seconds (1), 2 hours (3) and also classifiers that STEP fails to compile due to reaching the memory (MO) or time (TO) limits. The last two bars represent 19 and 9 classifiers, respectively. Finally, Figure 4c summarizes the performance comparison between XPXLC and STEP. In this comparison, the SDD compilation time is *ignored*, and the plot shows only instances for the classifiers that STEP is able to compile within the 2 hour time limit. Also note that both tools finish complete enumeration of PI-explanations for each of these instances. A point $(x, y)$ in the plot represents the time (in seconds) spent by XPXLC (shown on the $X$-axis) and by STEP (shown on the $Y$-axis) for a concrete data instance. Observe that, even if the compilation time is ignored, STEP's enumeration phase is still between 4 and 20 times slower than XPXLC.

**Assessing heuristic approaches.** Exhaustive enumeration of PI-explanations can serve to assess heuristic explanations. Exhaustive enumeration provides a distribution of how many times feature-value pairs appear in explanations, and thus which are likely to be more *relevant* for the given prediction. As a result, one can evaluate how many features in a heuristic explanation "hit" the set of most relevant (commonly-occurring) features. This strategy may be beneficial in some practical settings where trustable explanations are of concern. While our "hit" metric is a heuristic evaluation measure to compare the quality of explanations, we demonstrate its usefulness experimentally. For example, our metric does show a strong correlation between features of heuristic explanations and common features that we identify via enumeration. Figure 5 (included in the supplementary materials of the extended version of the paper [20]) depicts the percentage of features in explanations of Anchor [30] and SHAP [18] "hitting" the set of common features. Here, we focus on 2 datasets *Adult* [15, 30] and *Spambase* [37] and use the following methodology. For an explanation $E$ of Anchor, we keep the top $|E|$ features most commonly-occurring in all PI-explanations[15]; then we count the number of features in $E$ that hit the set of common features. As SHAP assigns numerical weights to *all* features, we take 5 features reported by SHAP as most relevant and count how many of them intersect the set of 5 most common features of PI-explanations. The rationale of this choice is that larger explanations are typically harder for a user to reason about and so 5 features is normally deemed enough to make a conclusion wrt. the cause of prediction. As can be observed, both Anchor and SHAP are successful at hitting the most common features. However, in some cases both tools' explanations do not overlap our important features, e.g. Anchor has zero overlap with the common features in more than 2000 instances. Given a significant overlap in the majority of cases, a zero hit suggests that Anchor's explanation might be using less influential features and is hence less trustworthy. This experiment illustrates another setting where PI-explanations can be useful, i.e. not only to output a provably correct explanation but also to provide the user with an alternative evaluation toolkit to measure confidence in heuristic explanations. Finally, we observe that both Anchor and SHAP are significantly slower than XPXLC: on average, Anchor takes 1.55 seconds to compute one explanation of an instance, whereas SHAP takes 99.58 seconds. In contrast, as highlighted above, XPXLC never exceeds a few tens of $\mu$sec for computing a single explanation.

## 5 Conclusions

This paper presents a log-linear algorithm for computing a smallest PI-explanation of linear classifiers. Moreover, the paper shows that PI-explanations for linear classifiers can be enumerated with polynomial delay. The results in the paper also apply to NBCs (among other classifiers), and so should be contrasted with earlier work [35], which proposes a worst-case exponential time and space solution for computing PI-explanations of NBCs. A natural line of research is to investigate extensions of XLCs that also admit polynomial time algorithms for computing PI-explanations.

## Broader Impact

Recent advances in the power of machine learning have not always been accompanied with the explainability of decisions made by complex models. The capacity to produce human-understandable explanations is sometimes not only an advantage but a legal obligation.

Linear classifiers, including the Naive Bayes Classifier (NBC), are ubiquitous in Machine Learning, being extensively studied and finding a wide range of practical uses. Recent work [35, 6] proposed worst-case exponential time and space algorithms for computing PI-explanations of NBCs, where PI-explanations denote minimal sets of feature-value pairs that are sufficient for the prediction.

Our paper investigates PI-explanations for linear classifiers, and proposes efficient (i.e. polynomial time) algorithms for computing one PI-explanation, but also efficient (i.e. polynomial delay) algorithms for enumerating PI-explanations. In practice, and for the specific case of NBCs, the new algorithms achieve orders of magnitude speed-ups over earlier work. More importantly, our algorithms enable computing PI-explanations for classifiers that were until now beyond the reach of existing approaches.

## Acknowledgments and Disclosure of Funding

This work is supported by the AI Interdisciplinary Institute ANITI, funded by the French program "Investing for the Future - PIA3" under Grant agreement n$^o$ ANR-19-PI3A-0004. M. Cooper is also funded by the project ANR-18-CE40-0011.

## Footnotes

[1]There is a growing body of work on explaining ML models. Example recent overviews include [9, 31, 32, 23, 22, 1, 24, 41, 25].

[2]Earlier work imposed the additional restriction of considering boolean-valued features. Clearly, non-boolean features can be binarized, e.g. with the one hot encoding, at the cost of adding additional features.

[3]In fact, the paper considers a generalization of linear classifiers, that accommodates both real-valued and categorical features, which serves to streamline the presentation. This generalization will be referred to as an *eXtended Linear Classifier* (XLC).

[4]It should be noted that for linear classifiers (including NBCs), heuristic explanation approaches based on linear approximations, such as those provided by LIME [29] or SHAP [18], can be regarded as uninteresting, since the model is itself linear. Nevertheless, aiming for coverage, we opt to include also results for SHAP.

[5] This section follows [27] throughout. An alternative would be to use Laplace smoothing [19].

[6] This example of an NBC is adapted from [3, Ch.10], with some of the conditional probabilities changed.

[7]Given the proposed reductions, it is immediate to represent an NBC with categorical features as an XLC.

[8]Pseudo-polynomial time algorithms for the knapsack problem are well-known [5, 26]. One concrete example [26] yields a polynomial (cubic) time algorithm in the setting of computing a smallest PI-explanation of an XLC. We show that it is possible to devise a more efficient solution.

[9] Proof included in the supplementary materials of the extended version of the paper [20].

[10] For a knapsack constraint, it is known that feasible solutions can be enumerated with quadratic delay [17, 12]. Nevertheless, we exploit the problem's special structure to achieve a log-linear enumeration delay.

[11] Proof included in the supplementary materials.

[12]The source code of XPXLC as well as the datasets, a demo and accompanying documentation are available at `https://github.com/jpmarquessilva/expxlc`.

[13]The CategoricalNB classifier of scikit-learn [33] was used for this purpose.

[14]This solution is not ideal, since the use of OHE impacts the assumption of feature independence of NBCs, and only serves to enable the comparison between STEP and XPXLC.

[15] If $> |E|$ features are in the top due to having the same frequency, all of them are marked as common. Also, the experiment is performed only for instances for which complete PI-explanation enumeration finishes.

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
