[Supplementary Material]

# A Supplementary Material

## A.1 Proofs

**Proposition 1.** *Let $\langle l_1, \ldots, l_n \rangle$ represent indices $\mathcal{E}$ sorted by non-increasing value of $\delta_j$. Pick $k$ such that $\sum_{j \in \{l_1, \ldots, l_k\}} \delta_j > \Phi$ and $\sum_{j \in \{l_1, \ldots, l_{k-1}\}} \delta_j \leq \Phi$. Then (12) holds for $\mathcal{P} = \{l_r | 1 \leq r \leq k\}$, and $\mathcal{P}$ represents an optimal solution of (16).*

*Proof.* We prove that an optimal solution to (16) can be obtained with the greedy algorithm that picks features in non-increasing order of $\delta_j$'s. Let $\mathcal{P}^* = \langle i_1, \ldots, i_k \rangle$ denote the $k$ indices in some optimal solution, such that $\delta_{i_1} \geq \ldots \geq \delta_{i_k}$. Moreover, let $\mathbb{V}(\mathcal{P}^*) = \sum_{j \in \{i_1, \ldots i_k\}} \delta_j$. Clearly, $\mathbb{V}(\mathcal{P}^*) > \Phi$; otherwise $\mathcal{P}^*$ would not satisfy the constraint in (16).

We prove by induction that one can construct another optimal solution $\mathcal{P} = \langle l_1, \ldots, l_k \rangle$, where $l_1, \ldots, l_k$ denote the first $k$ features with highest $\delta_j$. For the base case, we consider the first pick, and suppose that $i_1 \neq l_1$ (and so $l_1$ does not occur in $\mathcal{P}^*$). We can construct another sequence $\mathcal{P}' = \langle l_1, i_2, \ldots, i_k \rangle$, such that $\mathbb{V}(\mathcal{P}') = \sum_{j \in \{l_1, i_2, \ldots, i_k\}} \delta_j \geq \mathbb{V}(\mathcal{P}^*) > \Phi$. Hence, $\mathcal{P}'$ is still an optimal solution, and starts with a greedy choice. For the general case, we assume that the first $r-1$ picks can be made to respect the greedy choice, and that the $r^{\text{th}}$ does not. The reasoning now can be mimicked again, and so we can construct another optimal solution such that the $r^{\text{th}}$ choice is also greedy. Thus, Proposition 1 yields a smallest PI-explanation. $\square$

**Proposition 2.** *PI-explanations of an XLC can be enumerated with log-linear delay.*

*Proof.* For simplicity of presentation, we assume that the values $\delta_i$ are sorted in non-increasing order, i.e. $\delta_1 \geq \ldots \geq \delta_n$. This sorting operation can be achieved in log-linear time. Recall that $\delta_i \geq 0$ ($i = 1, \ldots, n$) and that a PI-explanation represented by the bit vector $p$ must satisfy the two constraints: (C1) $\sum_{i=1}^n \delta_i p_i > \Phi$ and (C2) $\forall j \in \{1, \ldots, n\}$ such that $p_j = 1$, $(\sum_{i=1}^n \delta_i p_i) - \delta_j p_j \leq \Phi$ (subset-minimality).

Consider an exhaustive depth-first binary search (DFS) in which at depth $r$ the two branches correspond to $p_r = 1$ and $p_r = 0$. It is critical for the correctness of this search that on each branch, the $p_i$ variables are instantiated in non-increasing order of the corresponding values $\delta_i$. For a depth-$r$ node $\alpha$ of this search tree, let $S_\alpha$ be the sum $\sum_{i=1}^r \delta_i p_i$. A node $\alpha$ is declared a leaf (and is hence not expanded) if $S_\alpha > \Phi$. Assuming that, by default, the remaining values $\delta_{r+1}, \ldots, \delta_n$ are assigned 0, node $\alpha$ satisfies (C1). Clearly, any other descendant nodes (at which at least one of $\delta_{r+1}, \ldots, \delta_n$ is 1) would not satisfy (C2) and hence does not need to be considered. This means that all PI-explanations will be found. It remains to show that all leaves $\alpha$ satisfy subset-minimality and hence are PI-explanations. To see that $\alpha$ satisfies (C2), let $\beta$ be its parent node. Since $\beta$ is not a leaf, we must have $S_\beta = S_\alpha - \delta_r p_r \leq \Phi$. But then $S_\alpha - \delta_j p_j \leq \Phi$ for all $j$ such that $p_j = 1$ since $\delta_j \geq \delta_r$ ($j = 1, \ldots, r-1$). Thus, all leaves correspond to PI-explanations.

We add to our DFS the pruning rule that a depth-$r$ node $\alpha$ is only created if $S_\alpha + \sum_{i=r+1}^n \delta_i > \Phi$. This sum is calculated incrementally, so only requires $O(1)$ time at each node. The reason behind this rule is that if it is not satisfied, then no descendant of $\alpha$ can satisfy (C1). On the other hand, if this rule is satisfied then we know that at least one descendant of $\alpha$ will be a leaf (and as explained above will correspond to a PI-explanation). It is well known that a depth-first search in a search tree with no dead-end nodes provides a polynomial delay algorithm [4]. In our DFS, the delay between visiting two leaves is linear in $n$. Since finding the first PI-explanation also requires a sorting step, with a log-linear complexity, we can conclude that the worst-case delay is log-linear. $\square$

## A.2 Additional Plots

(a) Anchor

(b) SHAP

Figure 5: Percentage of important "hits" of explanations produced by Anchor and SHAP.