[Reviews · NeurIPS 2020]

Review 1

Summary and Contributions: This paper shows proposes an algorithm that enumerates "PI-explanations" from a naive Bayes classifier, where the 1st best explanation can be found in polynomial time, and where each next-best explanation can be found with some additional polynomial time. Initially, I would have guessed that enumerating such explanations would be NP-hard, but after reading this paper, it is now obviously easy.

Strengths: This result was initially surprising to me, but now in retrospect it is obvious---this will really shift the way I think about this problem in the future, and likely some others as well.

Weaknesses: I expect some more practically-minded people will dismiss this work since it only (yet) deals with linear classifiers.

Correctness: The presentation is very thorough and clear, to the point where I believe the results are "obviously true".

Clarity: Yes

Relation to Prior Work: Yes

Reproducibility: Yes

Additional Feedback: Given that it is NP-hard to compile the decision function of a linear classifier [29], and given that finding the smallest prime implicant is Sigma_2^p hard, that enumerating smallest prime implicants of a linear classifier would still be NP-hard. But, now, it is obviously easy. Another, possibly-related surprise about this problem (explaining linear classifiers) is discussed in this recent paper: On Tractable Representations of Binary Neural Networks Weijia Shi and Andy Shih and Adnan Darwiche and Arthur Choi in KR'20 In particular, they note a folk result from learning theory, that implies that a linear classifier with integer weights can be compiled into an OBDD in pseudo-polynomial time. This was a surprise initially because again, it is NP-hard in general [29], but also now obvious in retrospect. For me, this paper provides an interesting result, but as it is, I admit that it may currently be of somewhat narrow interest. If there were broader implications, say in terms of explaining neural networks, I believe I could more strongly argue for this paper. For example, the above paper by Shi et al. applied this algorithm for compiling linear classifiers towards the compilation of some non-trivial neural networks, with some compelling analyses of some neural networks learned from data, provided through case studies.


Review 2

Summary and Contributions: The paper studies linear classifiers, in particular Naive Bayes. The focus is on explanations that can be computed in poly-time.

Strengths: 1. Bayesian networks, including Naive Bayes, is significant and has obvious relevance to the NeurIPS community. 2. The paper contains interesting theoretical and experimental results.

Weaknesses: 1. While the experimental section in the paper is nice, it could be improved with some more details, please see below. 2. The paper has both a quite broad focus (on explanations in AI, for black boxes, etc.) and narrow focus (on explanations for Naive Bayes and linear classifiers). There is substantial related work in the area of explanations in Bayesian networks that is not considered. Please see below for further information about this.

Correctness: In general, the claims and method appear correct. But I have a few questions about the description of «Reducing linear classifiers to XLCs» in lines 101 to 107: Where is this entering into the overall picture of explanations in this paper? For example, do you use this reduction in the theoretical part of the paper, in the experimental part, or both? Giving some more details on the role of this description would be helpful. The experimental evaluation in section 4 is reasonable, using the 37 datasets and comparing with the STEP compilation-based approach. It would be helpful to see some more details (number of features and instances, type of data,…) for the 37 datasets in the paper, instead of being referred to an online repository for all specifics.

Clarity: In general, the paper is well-written. This is a more detailed comment: the integration of equations and mathematical expressions into the running text is a little strange. For example, with respect to equation (9), you would want to: have a «,» right after the «)» of the equation and before «(9)», get rid of the comma in «Let,», add a word before «\sigma» and say for example «where \sigma». A similar comment applies to many of the other equations, where you want to put a period or a comma after the mathematical expression or equation, not before. And then start the next line after the mathematical expression or equation accordingly, with either a new sentence or a connecting word (like the «where» above) respectively.

Relation to Prior Work: There is some room for improved discussion of prior work, please see below. Different concepts related to explanations and Bayesian networks have been studied for several decades in the literature, see below for some example references: M. P. Wellman and M. Henrion. 1993. Explaining “Explaining Away.” IEEE Trans. Pattern Anal. Mach. Intell. 15, 3 (March 1993), 287–292. C. Yuan, H. Lim, and T. C. Lu. Most relevant explanation in Bayesian networks. Journal of Artificial Intelligence Research 42, 309-352, 2011. O. J. Mengshoel, D. Roth, and D. C. Wilkins. Portfolios in Stochastic Local Search: Efficiently Computing Most Probable Explanations in Bayesian Networks. Journal of Automated Reasoning, volume 46, 103–160, 2011. J. Kwisthout. Most frugal explanations in Bayesian networks. Artificial Intelligence, Volume 218, 56-73, 2015. C. S. Vlek, H. Prakken, S. Renooij, and B. Verheij. A method for explaining Bayesian networks for legal evidence with scenarios. Artificial Intelligence and Law volume 24, 285–324, 2016. With exception of Park's 2002 paper (reference [22]) there is no reference to the "explanations in Bayesian network" literature as sampled above. Clearly, the Naive Bayes setting considered in this paper is a restricted class of Bayesian networks. However, it would strengthen the present paper if there was an improved positioning of the work within the Bayesian network context, including how the paper differs from previous contributions such as the ones above (and related references).

Reproducibility: Yes

Additional Feedback: === Feedback on Aug 17 after opportunity for response from authors === I have read the response from the authors, and I think it is good. I believe the authors will be able to update the paper, considering the comments by myself and other reviewers, within the required timeline. All things considered, I remain positive about the paper, and in fact have bumped up its score one level.


Review 3

Summary and Contributions: This paper proposes a log-linear time method to compute the smallest PI-explanation for linear classifiers including Naive Bayes classifiers, improving the worst-case exponential time complexity of the compilation approach. It also shows that PI-explanations for a given instance can be enumerated with a log-linear delay.

Strengths: The proposed algorithm for PI-explanation significantly improves the time and space complexity of the state-of-the-art compilation based algorithm which has worst-case exponential complexity (although it is not limited to Naive Bayes or linear classifiers). Empirical evaluation also demonstrates this speed-up. The approach is also simple and natural: computing the smallest PI-explanation reduces to sorting the variables by their weights and choosing the first k, whether the classifier uses categorical, real-valued, or mixed features. This makes it easy to adopt and use this work.

Weaknesses: A major weakness of this work is that it is limited to linear classifiers, which are often already regarded as interpretable models in the explainable AI community. Moreover, the paper does not convincingly motivate and argue the benefits of explaining linear classifiers using PI-explanations. I think this is important because PI-explanations are computed directly from the ranking based on feature weights, which is how features of linear classifiers are often interpreted. On a similar note, as the experiments showed, there will be many PI-explanations (thousands to even millions) per instance, which will be hard to interpret.

Correctness: The algorithm for finding PI-explanation appears to be correct; the backtrack search approach for enumeration makes sense although I did not check the proof for log-linear delay in detail. The experimental comparisons are sound and thorough.

Clarity: The paper is fairly readable, but there is some room for improvement. After introducing extended linear classifiers which admit categorical or real-valued features (or both), restricting back to categorical case to describe the algorithm was awkward. The algorithm could be more cleanly described in the general XLC formulation. Also, the backtrack step of the enumeration was not very clear, in particular how it guarantees whether more PI-explanations exist. An example of the search procedure may help.

Relation to Prior Work: The paper describes PI-explanations in contrast with other ways of explaining ML models, and makes its algorithmic contribution clear against prior work on computing PI-explanations.

Reproducibility: Yes

Additional Feedback: In a sense, the feature weights are also a heuristic. How does ranking of features based on weights compared to the heuristics approaches (Anchor and SHAP)? In section 3.2, I would suggest changing the notation for value \Gamma to depend on instance e=a either in the subscript or as an input, to make it clear its dependence on a particular instance unlike the worst case value \Gamma^{\omega}. Also, additional notations like \Phi for -\Gamma feel superfluous and may even hurt readability. --------------------------------------Post-Rebuttal-------------------------------------- I have read the author response, and most of my concerns were addressed; I have increased my score by 1.


Review 4

Summary and Contributions: The authors describe how to compute cardinality-minimal / PI-explanations for the special case of linear classifiers. They show how to compute the shortest explanation, and how to enumerate explanations with polynomial time delay. Prior works rely on an offline compilation phase which may take exponential time in the worst case, or rely on calls to a SAT-solver (although can handle more general classifiers)

Strengths: The paper presents an enumeration algorithm for explanations of naive bayes classifier. The backtracking mechanism is tweaked so that the search will always find the next explanation with polynomial delay. It is interesting to see it laid out in detail.

Weaknesses: I'm not sure if the computation of a single explanation for naive bayes classifier is that entirely novel or interesting. It seems rather straightforward to greedily flip the least impactful variables. The main weakness of the paper is that the scope of the contribution is quite small. Linear classifiers are far from opaque, and I can't see a great need to generate explanations for their decisions. Even if practitioners do have the need to identify the minimal subset of features to guarantee a decision, I imagine most of them can immediately recognize the obvious approach of keeping the most impactful features. As such, I am doubtful that this contribution will be interesting to the NeurIPS community.

Correctness: Yes to the best of my judgment.

Clarity: Yes.

Relation to Prior Work: Yes.

Reproducibility: Yes

Additional Feedback: It would be interesting to see a discussion of how this work lies in comparison to classes of knowledge bases that enable tractable abductive reasoning [1]. For example, is this result a special case of some known class/language? [1] What makes propositional abduction tractable. Gustav Nordh, Bruno Zanuttini. ********************************************* Post-Rebuttal ********************************************* I have read the rebuttal and the other reviews, and may reconsider my score. I just wanted to address the author's request for specific references "that might cast doubt on the novelty of our work". Sorry for not being more concrete, but here are some specific references. [1] Near-Shortest and K-Shortest Simple Paths. W. Matthew Carlyle, R. Kevin Wood [2] Finding the k Shortest Paths. David Eppstein The polynomial time enumeration algorithm proposed for Eq 16 is basically subset sum where we enumerate all subsets that sum less than some threshold. This is a special case of the problem solved in [1] that enumerates all shortest s.t. paths in a DAG less than some length threshold, using polynomial delay and space. Let d_1,...,d_n be the weights from Eq16 and define d_{n+1}=0. The the reduction can be seen by constructing a DAG with nodes [0,1,2,3,...,n-1,n,n+1] and having edges from i to j for all i < j, where the weight of edge (i,j) is d_j. Then all simple paths from 0 to n+1 with length < k are subset sums with sum < k. [2] may also relevant where the same reduction can be used to enumerate subset sums in increasing order of sums, but I think it is not polynomial space. Edit: I've decided to bump by score up, since I realized the shortest paths algorithm does not satisfy the subset minimiality constraint that the author's method guarantees, so the author's enumeration algorithm is more novel than I originally thought.

[Author Response · NeurIPS 2020]

**Common comments.** Some reviewers raised the issue of the paper's *"narrow interest"* given its *"focused contribution"*, and due to linear classifiers (LCs) being *"interpretable"*. We cannot agree with the reviewers. First, LCs and NBCs are extensively used in different settings, with NBCs being deemed by some as one of the top algorithms in data mining. Second, the best-known heuristic explainers identify simple (local) linear models as a way of explaining complex ML models. Our work can be used jointly with these heuristic explainers for computing (local) PI-explanations. For instance, heuristic explainers can be used to produce more complex (and more rigorous) linear models, with more features, from which PI-explanations can then be computed. Clearly, one can envision other research directions relating heuristic and rigorous explanations. Third, interpretability depends on the number of features one needs to account for.

**Review #1.** *Q2:* We are very happy to read the reviewer's comment : *"this will really shift the way I think about this problem, and ..."*. We believe our results will impact a broader community if the paper is accepted.
*Q3:* It should be underscored that linear models are used for explaining complex ML models. Until our work, there was no efficient solution for computing a PI-explanation in those cases, and now there is one.
*Q8:* We will include a reference to the KR'20 paper. Thanks for the reference. However, that paper was not available when our paper was submitted. There was an earlier CoRR report, but it is from April 2020. The following paragraph merits comments: *"For me, this paper provides an interesting result, ... For example, the above paper by Shih et al. applied this algorithm for compiling linear classifiers towards ... "*. (i) We see a direct connection with neural networks and other black-box ML models. Well-known heuristic explainers find local linear models given a complex ML model and an instance. In those cases, we can compute PI-explanations. An efficient algorithm for computing PI-explanations of linear models will likely motivate additional results. (ii) The work of Shih et al. started in 2018 (IJCAI 2018) and focused solely on NBCs and monotone BNCs, with experimental results only for NBCs. The work has since been extended, but it started with a worst-case exponential in time and space approach for explaining NBCs.

**Review #2.** *Q3:* We will cover the references mentioned by the reviewer. These do not affect the novelty of our results, but are of course important for completeness. Thanks for the references.
*Q4:* We reduce NBCs to XLCs so that we can develop a unified algorithmic solution. The results are for XLCs but, given the transformation, apply to NBCs as well. Regarding the experiments, the main issue was the available space. We will include further additional detail in the supplementary materials, which already include additional detail.
*Q5:* We will address the comments on notation.
*Q6:* The references will be added and we will relate those with our work.

**Review #3.** *Q3:* The following statement merits comments: *"On a similar note, ..., there will be many PI-explanations (thousands to even millions) per instance, which ..."*. First, we showed these results solely to demonstrate that our algorithms scale to large problems, with many features and with many explanations. Second, analysis by hand of millions of PI-explanations is unrealistic. However, such PI-explanations can be analyzed automatically, e.g. to gather statistics and test hypotheses. Also, we can enumerate PI-explanations that respect some additional properties. There are many possible scenarios, as long as enumeration of PI-explanations is easy, which it now is.
*Q5:* We opted for a restricted version of XLC to describe the algorithms to keep the notational overhead to a minimum. Also, all proofs are included in the supplementary materials.
*Q8:* The reviewer is correct that weights in linear classifiers are heuristic. However, that is orthogonal to our work. If one fixes the linear model we will compute rigorous PI-explanation in log-linear time. If a different (extended) linear model is picked, the same rationale applies. Heuristic explainers are significantly different than what we propose. Heuristic explainers either approximate a complex ML model with a linear model (in the case of LIME) or heuristically identify a set of literals as a tentative explanation (in the case of Anchor). As stated above, our approach can be used with LIME or SHAP, or other heuristic explainers that identify a linear model. And this extends significantly the reach of our work, but also the reach of earlier work. We will address all comments regarding readability.

**Review #4.** *Q3:* The review is not entirely correct since our algorithm is not based on *"greedily flip the least impactful variables"*. No variables are flipped; flipping is only used for implementing backtracking. Our algorithm works in such a way that entailment is guaranteed at each step, this without directly checking entailment with an NP oracle. This is novel and likely to change the way these problems are looked at (as noted by Reviewer #1). Before our work, the best only approach for computing PI-explanations of NBCs was given in [29], but this is worst-case exponential. The approach hinted by the comment *"keeping the most impactful features to guarantee"* is insufficient, and would require additional reasoning to ensure that entailment is preserved. This is what our algorithm does, and that reflects part of the novelty. Also, our algorithm enumerates explanations with log-linear delay. No heuristic approach for computing explanations is capable of deterministically enumerating explanations. We kindly ask the reviewer for references that might cast doubt on the novelty of our work. Otherwise, we would expect the reviewer to revise their scores.
*Q8:* Our results do not follow from the work of Nordh and Zanuttini since a linear classifier is a global function employing real coefficients whereas their work considers language-tractability of purely logical functions.

[Meta-Review · NeurIPS 2020]

This paper makes a narrow but solid contribution towards explaining linear classifiers.